# Study on the Effects of Different Cutting Angles on the End-Milling of Wire and Arc Additive Manufacturing Inconel 718 Workpieces

**DOI:** 10.3390/ma15062190

**Published:** 2022-03-16

**Authors:** Gustavo Quadra Vieira dos Santos, Jun’ichi Kaneko, Takeyuki Abe

**Affiliations:** Graduate School of Science and Engineering, Saitama University, 255, Saitama 338-8570, Japan; jkaneko@mech.saitama-u.ac.jp (J.K.); abe@mech.saitama-u.ac.jp (T.A.)

**Keywords:** additive manufacturing, arc welding, machining, cutting angle, cutting forces

## Abstract

This research presents an analysis of the effects of different cutting angles on the side milling of Inconel 718 products manufactured with the Wire and Arc Additive Manufacturing (WAAM) technique. Considering that this manufacturing technology can build near-net shape products, its surface quality is deemed unqualified as a final product, requiring a post-processing step. In this paper, three different angles—0°, 35°, and 90—are compared, looking for possible differences regarding its machinability. As the alloy in question is a material known for being difficult to machine, and the samples were produced with the additive manufacturing technique that created peculiar characteristics, it was deemed necessary to analyze different aspects of the machining process: the surface quality, tool wear, and cutting forces for all three cases, and to rank the angles regarding these results. With analog experiments with the same alloy but cold-rolled, it was possible to infer that not only is the 0-degree angle is the best option for milling, but the anisotropy of the WAAM samples could be the major source of the differences in the milling results.

## 1. Introduction

Recently, there has been a growing trend in the adoption of the different Additive Manufacturing (AM) techniques in industry and the academy. This increase in the utilization of AM has translated into the growth of collaborative research, translation of technology, and its commercialization on a global scale [1].

Although the use of AM might be more known to the public for its applications based on polymers, metal additive manufacturing is considered a fast-growing part of this industry [2].

Considering that the adoption of AM techniques introduce new kinds of unknown factors to the production cycle, and, for a better understanding of these factors, and consequently for better control of the manufacturing process itself, it was deemed necessary to understand how the different steps of the product fabrication could be influenced by the introduction of this relatively new way of manufacturing.

As most AM techniques share common features, each of them has its peculiarities, and due to these differences, the approach for posterior manufacturing steps might change. Among those methods, the techniques that fall under the definition of the Direct Energy deposition (DED) processes are quoted as being the most suited for the manufacturing of larger components with medium complexity [3].

One technology that belongs to the DED category is the Wire and Arc Additive Manufacturing (WAAM) technique that could be defined as a process for building medium to large near-net shape products utilizing wire as feed material and arc welding as a heat source [4]. This technique has attracted attention for its metal building capabilities and its high efficiency, and relatively lower costs [5]. 

In contrast to the previously mentioned advantages, due to the deformations caused by high heat input, the quality of the surfaces being manufactured with WAAM is not suitable for most final products; with that, a post-processing step is required [6]. Notwithstanding, even with the difficulties of manufacturing with a relatively novel technique, new examples of products being produced with the WAAM method are surging, ranging from marine propellers [7] and aeronautical structural panels [8] to rocket engine combustion systems [9]. Based on the type of use and consequently necessary required properties of materials like aluminum, copper, titanium and nickel alloys are employed for metal additive manufacturing.

Among the materials used for AM, due to some exceptional properties such as strong corrosion resistance, toughness, and strength at high temperatures, the Inconel 718, a nickel-based superalloy, is highly sought after. It is widely used in the aerospace industry and fields that require these specific features [10]. Although it has several advantageous properties, this alloy also comes with some hindrances. One of these problems is its challenges in terms of machining [11], explained by the short tool life owing to the abrasion and work hardening processes and the high cutting forces, which can possibly incur damages during the machining process [12].

On the other hand, this kind of limitation and the high cost of material acquisition for the Inconel 718 alloy are characteristics of a material that could benefit from producing components with the AM manufacturing process [13], reducing the machining time and eventual material waste.

While the use of WAAM to produce parts based on expensive alloys is quoted as being a possible improvement for the manufacturing process, the introduction of a new technique could also change the properties of its products. In the case of the Inconel 718, modifying characteristics such as microstructure, tensile strength, elongation, and yield strength [14], can possibly affect other manufacturing steps.

Regarding the microstructure, and consequently, its properties involving this kind of technology, based around the melting and deposition of material into multiple layers with a rapid solidification rate, this different kind of thermal history ends up influencing the formation of its microstructure. Thus, just like a typical welding process, when supercooling occurs, the microstructure formed is composed of dendrites [15]. This is not the only difference found in WAAM products, but these products also present a high degree of anisotropy [16].

Taking into consideration the process capability of producing near net shape products with medium complexity, it was deemed that when machining this kind of part, different tool paths, and consequently, different cutting angles could be chosen. Examples of this kind of process have been increasing. In addition, the amount of research regarding hybrid AM processes using not only regular machining centers but also robots [17,18,19,20] is increasing.

With the idea that different mechanical properties are generated by the additive manufacturing process in mind, the machinability of Inconel 718 samples and mainly the influence of different machining parameters in this kind of alloy should be further evaluated.

This paper presents a study of the outcomes of side-milling with different cutting angles of Inconel 718 workpieces produced with the WAAM technique. In the next section, the different parameters of both the manufacturing process, the machining procedure, and data acquisition will be discussed.

## 2. Materials and Methods

Before the actual milling procedure, samples were printed using a Manipulator Robot (Arc Mate 100iC, Fanuc, Yamanashi, Japan) with MIG welder equipment containing a wire feed unit (VR 7000 CMT, Fronius, Pettenbach, Austria). Walls were built with 4 mm of depth, 30 mm of length, and 20 mm of height. These specifications were chosen based on different aspects of the experiment procedure, for example, the machining data acquisition process. The printing parameters were fixed, including printing direction, interlayer deposition delay, current, voltage, torch, and wire speeds, and are present in Table 1.

It was defined that the machining process would be a side-milling procedure. Different reasons were behind the choice for the milling process. Apart from its growing popularity as a part of a hybrid process of AM and subtractive techniques, as quoted previously, not only the end-milling is one of the most employed metal cutting procedures in general [21], but also this kind of metal removal process is the usual choice because machined samples are composed of thin walls [22].

Considering the type of material, it was also chosen to perform a down-milling operation since it was confirmed to have advantages like longer tool life [23] and better surface finish [24] when milled with the Inconel 718 alloy. The milling process was specified to be a dry-milling operation and to observe the effects and characteristics of the different cutting angles along with the feasibility of the machining without the aid of cutting fluids.

For the analysis of the influence of the material deposition direction on the machinability of Inconel 718 workpieces manufactured with the WAAM technique, different steps were performed, and setups were necessary. First, during the machining process itself, the cutting forces were acquired using a stationary dynamometer (Type 9272, Kistler, Winterthur, Switzerland), coupled in a Vertical Machining center (VM4 Machining Center, OKK Corporation, Osaka, Japan).

For this data acquisition step, the samples were set up in the Dynamometer having differences in tool posture. Figure 1 contains three pictures of the workpiece set up with the different machining angles, 0 degrees (case a), 35 degrees (case b), and 90 degrees (case c). For an easier understanding of the relationship between tool posture and deposited layers, Figure 2 was drawn.

Digital and high-speed cameras were set up inside the chamber, the former, for the observation of possible disturbances in the cutting process.

For the experiments, only the cutting angles presented above varied. The other cutting parameters were fixed, having a feed rate of 120 mm/min, spindle rate of 800 rpm, and cutting speed of 15 m/min. Three different cutting tools of the same type (WXL 3D DE, OSG Corporation, Toyokawa, Japan) were used, being them, two-flute coated carbide endmills, with 6 mm of diameter and 35 degrees of helix angle.

It was also designated that 0.3 mm of radial and 5 mm of axial cutting depths were performed in three different heights, that is, at five, ten, and fifteen millimeters from the top of the samples, with a different number of machining, passes at each height. Being the count of passes were 9, 8, and 7 consecutively, and with that, having a depth difference of 0.3 mm for each height, the final samples are presented in a staircase format for clarity. The machining parameters quoted in this section are present in Table 2.

Apart from the chosen parameters, some additional explanations were deemed necessary since there are peculiarities regarding the additive manufacturing samples that could affect the machining process and are present below.

Due to the different characteristics of the AM samples, even though there was an attempt to make the experiments as equal as possible, some differences could be noted. Owing to not manufacturing parameters that were the same for all samples, but technique characteristics like the unevenness due to the high heat input and the stair-stepping effect that occurs between the deposition of the two layers. As a result of this problem, degradation of the surface quality [25] occurred with width and height varying across the sample.

Regarding those differences, the sample heights presented an average variation of around 3 mm (minimum average of 22.17 mm and a maximum of 25.16 mm) and width of around 1 mm (minimum average of 5.19 mm and maximum of 6.20 mm). With that and the angle differences, the amount of material removed varied between the different experiments.

Considering the above-mentioned information, two variations in the results could be observed, which are thoroughly discussed in the later sections of the present paper. The first variance that affected the results of cutting forces, being a difference that influenced the distinct machining passes of the same sample, is a variation that occurred due to the deformation of width, which was discussed previously, and occurred until there was a uniform removal of material, that is, the milling of a flat surface.

The other difference of the experiments did not affect the maximum cutting force values, but as it affected the length of cutting, it could mean a difference in the tool wear, but, due to the small variation in length and this distance overall, it could be considered negligible to the comparison of the results.

Due to the cutting parameters, mainly the angle difference, some variations could be observed, first between 0 and 90 degrees. Being the height of the samples is different when compared with the length, there is a small difference in cutting distance, varying of around 3 to 5 mm.

The other type of discrepancy occurred when machining with 35 degrees of inclination; due to this cutting angle, the length that was milled varied across the different cutting layers, increasing as it moved farther from the top of the sample, is this kind of difference not only expected but inevitable.

For easier visualization of the differences in height and variations on the samples, Figure 3 was assembled, which contains pictures of two different perspectives of the deposited samples.

As discussed previously, multiple cutting passes would be performed to assess the peculiarities of the additive manufacturing samples. One of these is the structure; having this characteristic brings the possibility of division into two different levels, the macro-and micro-level structures. On the macro level, the peculiarity of the AM sample, as opposed to other products manufactured with different methods, is seen with the naked eye, as stated previously, due to the manufacturing process and deformations caused by the heat input.

The other level quoted previously is linked to the microstructure of the WAAM samples. As explained in the introduction, the thermal history during deposition induces the formation of a dendritic microstructure, for the specific case of this research, composed of columnar dendrites. To exemplify the microstructure of the Inconel 718 WAAM products, a sample was manufactured using the same parameters as the samples used in the different steps of this research, and its microstructure was observed after being cut and posteriorly polished and etched. This example is presented in Figure 4.

To observe a possible influence of these microstructure peculiarities on the machining results, different passes were conducted after the surface reached a flat state. With that, the deformations on a macro level would not influence the machining process, leaving a possibility of further analyzing the microstructure aspect of the machining operation and the possible effects of the different cutting angles. For the first evaluation then, the quality of the post-processing step was analyzed and is present in the next section.

## 3. Results and Discussions

### 3.1. Machining Quality Assessment

To assess the quality of the machining process, a surface examination was done, composing this phase of visual observation, which was carried out with the aid of a microscope, evaluating the surface aspects, such as feed marks and possible surfaces imperfections. A surface roughness analysis (Ra and Rz), with the aid of a profilometer (ET200, Kosaka Laboratory Ltd., Tokyo, Japan), was carried out; with that, the surface quality the process stability were further analyzed.

As stated previously, the first part of this assessment consists of a visual observation of the samples. One example of this analysis is present in Figure 5, which is composed of the three pictures of the machined surfaces showing the initial results of the machining procedure.

Different aspects of the samples could be observed with the aid of the microscope. First, the feed marks are easily located in the samples and are more visible on the outer parts of the machined surfaces, especially on pictures (b) and (c). When further analyzing the feed marks, it is observed that they presented distinct characteristics for the different cases.

Case (a) had a regular and constant spacing of the marks throughout its surface, which did not occur with the other two cases. Although in some parts of the samples (b) and (c) a certain degree of regularity of the spacing between cut marks was visible, the most observed surface appearance is what seems like non-periodic marks.

The last two pictures present different irregularities not related to the cut marks, which were hypothesized as the effects of possible chatter vibrations or from plastic deformation. The reasons for these variances are further discussed in various parts of this paper as opposed to picture (a), which have a more even surface on the three different layers.

In terms of these irregularities, it is possible to rank the three workpieces, and posteriorly, these visual observations will be compared with the surface roughness results. Sample (b) presented more inconsistencies, (c) the second most irregular, and at last, (a) was the sample that presented the less uneven surface.

The different heights of the samples, however, be it for possible tool wear, different hardness, or even microstructure, do not appear to have a relation to the surface appearance, presenting about the same types of irregularities for the three analyzed cases.

After this initial step, to better evaluate the quality of the workpiece surfaces, as previously explained, the surface roughness was measured for the three different heights of each sample, according to the ISO 4288 standard for milled surfaces [26]. The values for Ra were assembled in Table 3, similarly to the values for Rz in Table 4.

Different observations can be made from the results seen in Table 3. Firstly, it is interesting to note that the values of Ra match the evaluation that was performed previously regarding the visual aspect of the samples. That is, when the perceived irregularities were ranked by the authors, the order of worse to best was (b), (c), and (a); equivalent results were found for the roughness values.

Another aspect of the experiments that could be inferred with the results from Table 3, which was hypothesized previously, i.e., that there is no apparent correlation between sample height, machining order, and surface quality. As tool wear occurred, which will be discussed later, it did not have a major effect on the surface quality of the same sample.

As opposed to the values of Ra, which are average values of the profiles measured, the Rz data are measures of the highest peak and valleys observed in the sample, and even though the (b) and (c) values are closer than previous values, the values for the workpieces machined with 0 degrees of cutting angles continue to be lower, indicating that not only on average it had fewer irregularities, but these differences were significantly smaller in size when compared with the other two cases.

As explained previously, the experiments were composed of different machining passes, nine for the first height, eight for the second height, and seven for the last height. Considering the dimensions of the deformations present in the samples, it could be observed that around the fifth cutting pass, the surface being cut was already flat, meaning that possible vibrations and irregularities that appeared in the final surfaces were not influenced directly by the deformations present in the workpieces, with that, two hypotheses were formulated to try and explain these differences, being them further discussed on the cutting forces analysis, and conclusions parts. To further examine the irregularities present on the surfaces after machining, the tool wear was observed, and this examination is discussed in the next topic, followed by an analysis of the cutting forces.

### 3.2. Tool Wear

Although the amount of material removed was considered small, and keeping in mind that the Inconel 718 is a material with low machinability, and the WAAM manufacturing method could influence the different mechanical properties of it, possibly making the material removal process more difficult, it was deemed necessary to conduct this tool wear analysis.

For the evaluation of the tool wear that occurred in the three cases, pictures were taken with a microscope in the same manner as the surface appearance evaluation and were analyzed regarding the different wear mechanisms. The different cases (a, b, and c) were ranked regarding the dimensions of wear; Figure 6 contains examples of the pictures taken of the three tools.

In all examined cases, the wear is highlighted by the red circles was flank wear, which occurred primarily via abrasion. This wear was also considered minor for all samples, ranging from 0.14 mm in length for the most impacted sample (c) and 0.10 mm for the less impacted one (b).

When considering that the surface roughness and appearance had not suffered a decrease in quality when comparing the different machined layers of the same workpiece, as quoted previously, that is, the order in which the heights were cut did not correlate with the decrease in surface quality, it is possible to infer that the tool wear did not have a major influence on this aspect of the machining process.

When further analyzing the tool wear mechanisms, an abrasion form of wear was observed in the cutting tools, as well as some degree of chipping was also visible on the blades, especially when it comes to the last machined sample (c). An image containing an example of the chipping, highlighted in red for the 90-degree angle case, is presented in Figure 7.

It was previously explained that when mounting the machining setup, to evaluate possible disturbances in the machining process, not only a digital camera but also a high-speed camera was installed inside the machining center chamber. Even though the digital camera did not manage to observe apparent instabilities in the machining process, the high-speed camera was able to capture some valuable data regarding the tool wear. Images representing these results for the three different cases (0, 35, and 90 degrees) are shown in Figure 8.

The valuable information that was mentioned previously is highlighted in the red circles of Figure 8 and happened similarly for all the cases studied. And with this additional knowledge, it is possible to comprehend some aspects of the wear present in the cutting tools.

When observing the machining procedure, it was possible to notice that some of the material that was being removed from the workpieces, the machined chips, were not expelled in a fast manner; they became welded to the cutting tools. Being removed only after a few rotations of the same, and the pictures in Figure 8 show one of those moments for each cutting angle when one chip was still turning along with the cutting tools.

The welding of the chips to the cutting tools represents an undesired effect of the machining process, usually happening due to the high temperatures that occurred during the experiments. This kind of effect could have a direct impact on the tool life since the welded chips, when leaving the cutting tools, end up removing small amounts of the tool’s material, being called adhesion wear, and this cumulative amount could represent significant wear when considering larger cut lengths. This kind of wear and the previously discussed abrasion—known problems of the machining of Inconel 718 workpieces— are also discussed by different authors [27,28,29].

In the next section of this research, the cutting forces are presented and analyzed, not only with the focus of a further evaluation of the difference of each cutting angle but also by discussing some distinct AM product characteristics.

### 3.3. Cutting Forces

As explained in previous sections, the cutting forces were acquired throughout the different experiments, and with that, distinct results could be gathered, and in this section, three major discussions will be presented. First, with the aid of Figure 9, which shows the data for the cutting forces for a single cutting cycle (full rotation of the cutting tool). The second, visualized in Figure 10, with data regarding one entire cutting pass, and the last discussion, pictured in Figure 11, the maximum forces for each cutting pass.

With the aid of Figure 9, the observation of the process stability is possible, and even though the first case, 0 degrees, shows some level of instability at the *x*-axis, as discussed in the previous discussion compared to the other two cases. The 35-degree cutting angle, in particular managed to have a more constant process.

From Figure 10, it can be seen that although the results for surface roughness and visual aspect, discussed in Section 3.1, Machining Quality Assessment, for the 0 degrees of cutting angle are superior in quality, the values for the three cases present a varying trend. At the same time, the values for the *y*-axis are slightly higher for the 35-degree angles. On the other hand, the *x*-axis values showed a significant difference, as shown in case (a), with higher values on average.

When it comes to the shape of the graphs of Figure 10, they were considered similar, and little to no conclusions could be drawn. Although some degree of variability is visible, especially for the *x*-axis in the case (a) and *y*-axis for (b) and (c), the instability was better visualized with Figure 9.

Figure 11, as explained previously, presents the maximum cutting forces obtained throughout all the different machining passes for the *x*- and *y*-axis for the three different angles discussed.

Figure 11 presents a few expected results; first, it is possible to observe a more even trend when it comes to the *x*-axis values, having the three cases similar results on average. The different angles presented a significant increase when comparing the first machining passes to the last one for the same height, indicating the growth of the amount of material removal, directly influencing the cutting forces. Another aspect to be discussed from the *x*-axis results is the fact that the maximum values, especially comparing the last machining pass for the different height, does not present a high amount of variation; this data contributes to the already discussed idea of a lack of significant tool wear, which could increase the cutting forces consecutively.

The *y*-axis does not present the same trends observed in the *x*-axis. First, the variability is higher when considering the 35- and 90-degrees cases, opposed to the upward trend observed in the previous case. Especially for the 35 degrees results, the maximum cutting forces do not seem to follow a tendency of increasing when observing the same height for these two angles.

The 0-degrees angle, on the other hand, presented a tendency like the one presented in the *x*-axis, having a more seemingly stable process. The observations of Figure 11 do not only match the results gathered previously for surface quality but also the results for cutting forces analyzed in Figure 9 and Figure 10, which complement the idea of a greater tendency of instability for the 35-degrees and 90-degrees angles.

When considering the stability results for this topic and the previous results, it was possible to hypothesize two possible reasons. Being the first possibility, the influence of the microstructure on the cutting procedure is discussed next.

When evaluating the material to be anisotropic, a consequence of the dendritic microstructure, although the values of cutting forces did not vary considerably, the influence of this characteristic in the cutting mechanism [30] is thought as being one of the reasons for the results presented in the present paper.

The anisotropy of additive manufacturing samples, not only for WAAM and the specific case for Inconel 718 alloy, was proved as being a major influence in the post-processing procedure, for not only cutting forces and surface quality [31] for different build and cut directions but also tool wear [32].

Another point to be considered regarding the influence of the microstructure on the machining results is the difference in the microstructure itself for the studied samples. Having this kind of microstructure, the presence of dendrite and interdendritic structures that could have different values for mechanical properties, in a similar manner as the stress–strain for single-crystal nickel alloys differ [33], possibly causing disturbances in the machining procedure.

To better illustrate this hypothesis, a representation of each machining procedure at the microscopic level was drawn and is presented in Figure 12 to show the differences for each cutting angle. For better comprehension of the microstructure of the samples in question, a microscopic picture representing the dendritic structure of the Inconel 718 WAAM as deposited at a higher magnification than the one in Figure 4 is shown in Figure 13.

With Figure 12 and Figure 13, some of the peculiarities of the dendritic microstructures can be explained more easily. First, when compared with regular grain structure, this kind of microstructure presents innate irregularities. It is possible to be observed that the surface, despite being polished, does not appear to be flat, having a clear contrast between the dendrites that appear as “peaks” and the rest of the surface.

The dendritic structures shown in Figure 13 have a different texture when compared with regular grain structure alloys. This kind of microstructure is highly anisotropic and could influence the mechanical properties’ anisotropy as well [34].

Another characteristic concerning this kind of structure is regarding its growth direction. The dendrites grow in the direction of the deposition, developing themselves across the deposited layers due to the remelting of the previously solidified layers [35]. In Figure 13, this deposition direction is shown as well as the indicator of the dendrite growth direction.

These differences in microstructure and the influence that the dendritic structures present, as explained previously, are hypothesized by the authors as being one of the possible causes for the diversity of results for the different cutting angles. Another difference regarding the microstructure that could represent a further explanation for the deviations in the cutting forces consists of the variations in diameter for the dendritic structure across the different axes. Although not observed by the authors in the present research, these differences are reported and will be further discussed in the present topic.

The second hypothesis that was formulated to explain the differences found in the results is based on the characteristics of the samples. The workpieces that were deposited and later machined were chosen to be walls of around 5 mm of width, and this characteristic could be detrimental to the manufacturing process.

It is known that thin walls are more susceptible to negative influences during the machining procedure, such as vibrations that can occur during the cutting steps and that can affect the surface finish [36] and cause tool deflections [37]. And even though there is a lack of study regarding the effects of the different cutting angles in this type of sample as well, one could assume that the effects of the differences in cutting forces directions could be amplified in this type of sample, being not a characteristic specific to the AM products, but of the sample’s dimensions.

Although the second hypothesis that was formulated was not completely ruled out by the authors, when considering the data gathered throughout the research, there was no evidence suggesting it was the real cause of the differences observed. When observing the cutting forces, there was no apparent sign of both chatter vibrations and tool deflections, meaning that the first hypothesis is the assumption most backed up by the data.

Even if data were indicating the first hypothesis to be a good representation of the effects of the tool postures in relation to the deposition direction, it was decided to perform further experiments that could reinforce or discard that assumption. This time, however, the samples to be machined were not AM samples but were composed of the Inconel 718 material, more specifically the AMS 5662, an alloy that went through both cold rolling and Solution heat treatment processes.

This type of Inconel 718 alloy was selected for being first, a material that was not age-hardened, presenting a hardness value that is closer to the AM samples presented in this paper, and most importantly, presenting both microstructure and mechanical properties that are isotropic. Apart from the anisotropy aspect, these two kinds of materials also present other different characteristics, and some of them will be presented next.

For instance, when it comes to the microstructure, it was already informed that the WAAM samples are composed of dendrite formations, this kind of microstructure has also been reported to have different diameters throughout the three axes, ranging from sizes as small as 53 µm to as big as 615 µm [9]. As stated previously, this could be another influence for the differences found in the cutting force values.

The AMS 5662 alloy, on the other hand, is composed of equiaxed grains [38], meaning the grains share about the same diameter. The requirement stated by the AMS standard, to be of around 63 µm or finer.

Regarding the microstructure, one important difference between the two kinds of samples is regarding the presence of Laves Phases. This intermetallic phase is reported as being not only brittle but is also linked with the loss of ductility for the Inconel 718 alloy [39]. While the WAAM Inconel 718 is known for having this kind of intermetallic phase deposits among the precipitates of its microstructure [40], the AMS 5662 does not have the presence of this phase.

While considering the previously mentioned differences, in the next topic then, different aspects regarding the machining of the AMS 5662 are presented.

### 3.4. Anisotropy Hypothesis Investigation

As mentioned previously, the objective of this research topic is to better analyze the causes of the differences in the machinability results among the different cutting angles obtained and discussed previously.

For that reason, by machining a sample known for having isotropic mechanical properties at different cutting angles, the data could be used to better understand the possible influence of the anisotropy of the WAAM samples on the machining process.

For this analysis, it was chosen to perform the milling steps following the same parameters as the other kind of sample, data that is present in the Section 2 and summarized in Table 2. However, in terms of the cutting angles, it was chosen that only two angles would be analyzed, the 0 and 35 degrees, to be referred to as cases (d) and (e), respectively. For the cases presented with the most differences among the three experiments, it was considered that the results concerning these two angles would be sufficient data for the objective of this paper.

In a similar manner to the discussion for the samples produced with the AM technique, evaluations regarding the surface quality and cutting forces for a single cycle will be presented in this section. The first analysis step addresses surface quality.

Figure 14 presents a comparison for both cases, showing pictures of the surfaces after the machining procedure, taken with the aid of a microscope.

As opposed to what was observed in Figure 5, the surfaces after machining present similar appearances, with the sample (e) appearing to have minor irregularities, being possibly plastic deformations due to both high temperatures generated during machining and the effect of tool wear. Differently to what was seen in the WAAM process, with major changes for the varying cases.

Following the same protocol adopted for the analysis of the WAAM samples, the next step for the surface quality evaluation would be the surface roughness measurements. This time the surface roughness average values (Ra and Rz) for cases (d) and (e) respectively were measured and are present in Table 5 and Table 6.

In Table 5 and Table 6, what was observed in Figure 14 becomes more evident, having the slightly more irregular appearance of the case (e) being matched with the differences in the surface roughness. Although these differences could be considered minor changes, of around 15%, especially when compared with the variances observed on Table 3 of roughly 56%, from the 0- and 35-degrees cases.

Even though not further addressed in this paper, it was deemed important to mention that the same kind of wear process was observed for the machining of the AMS 5662 sample, having not only the abrasive wear shown, but the wear mechanism seems to be unchanged. From high-speed camera footage, the welding of the machined chips to cutting tools was also observed, not being then a particular effect of the AM products.

As the final step of this paper, the cutting forces for a single cycle for both cases (d) and (e) were analyzed, following the same procedure that was performed for the previous sample type. The results were plotted in graphs and are present in Figure 15.

From Figure 15, different analyses could be made, first, regarding the values themselves. They are similar in magnitude, especially the *x*-axis and *z*-axis, and have a slightly lower value for the *y*-axis in the 0 degrees case.

Regarding the stability of the process, they seem to be also close, with the 35 degrees angle having seemingly more unstable values for the second turn of cutting (different blade), as opposed to a more stable process for the 0-degree case for the “three cuts.” This difference in stability was hypothesized as being a specific problem due to possibly an increase in the tool wear of the blade in question and not necessarily a trend for this type of cutting procedure.

Although the differences observed in this topic are not considered major differences, some degree of change occurred, and possibilities like instability due to the clamping system or due to the nature of the samples themselves (thin walls) are not discarded.

When comparing with the additive manufacturing samples, however, the changes in the results are at a smaller scale, indicating that even though that might not be the only cause of the different results in surface quality and cutting forces, the anisotropy of the AM samples is still the major hypothesis of the cause of differences for the machining of different angles for the wire and arc additive manufacturing Inconel 718 parts.

In the next topic, a conclusion of the results and a consideration for future works are presented.

## 4. Conclusions

(1)It was possible to observe with these different research characteristics of the machining of Inconel 718 WAAM samples, considering the deformations of the workpieces used by the high heat input and the possible influence of the microstructure on the machining of those samples.(2)As the last machining pass of each height was performed at a flat surface, it was possible to remove the possible influence of the deformations on these cutting passes. And, as the results for not only the cutting forces for a single cutting cycle differ but also the results for the surface quality are also unequal, the microstructure and its anisotropy probably played an important part in these results, directly influencing the cutting process and its results.(3)Although the cutting tools did not present much wear during the experiments, the primary cause of wear matched the mechanism discussed for the machining of wrought Inconel 718 by other authors, and the AMS 5662 was performed on the second step of this research.(4)Different aspects of the machining process for three different cutting angles of Inconel 718 workpieces produced with WAAM were evaluated and discussed. An exception for the tool wear process, which presented similar results for all the different cases, not only considering the visual aspect and surface roughness results, but also the cutting forces stability, it is safe to assume that the 0 degrees angle proved to be the optimum angle when comparing the three cases for the machining of Inconel 718 WAAM samples.(5)Regarding the reasoning for the different results hypothesized by the authors after the machining of AMS 5662 samples, although some differences in the results occurred, the possibility of the influence of anisotropy is still the main hypothesis for the major changes in results for the different cutting angles.(6)Even though some different results were acquired in the present research, the authors deemed it necessary to further investigate the possible influences of the additive manufacturing anisotropy and the dendritic microstructure on the machining process. Some of these influences include the additive manufacturing parameters and the machining parameters, such as cutting angle and types of cutting tools, considering not only “regular” machining processes but also the micro and precision machining of AM products.

## Figures and Tables

**Figure 1 materials-15-02190-f001:**
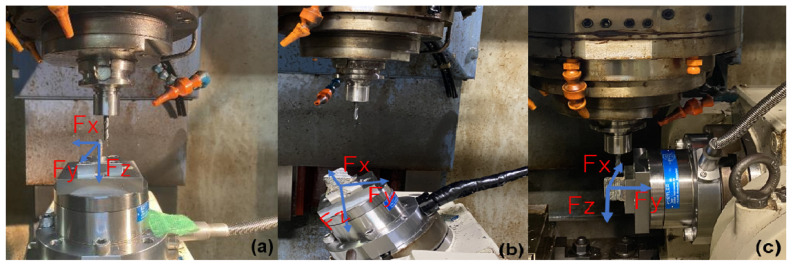
Machining Procedure Setup. Cuttings angles of (**a**) 0 degrees, (**b**) 35 degrees, and (**c**) 90 degrees.

**Figure 2 materials-15-02190-f002:**
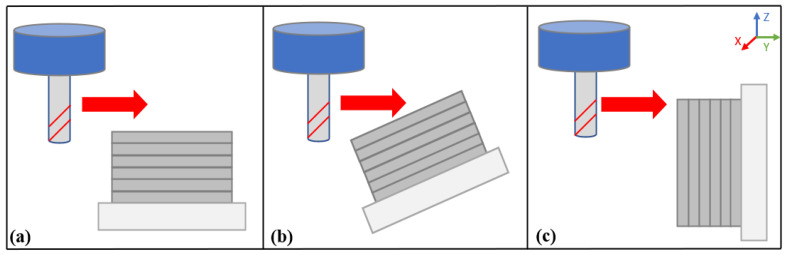
Tool posture representation—(**a**) 0 degrees, (**b**) 35 degrees, (**c**) 90 degrees.

**Figure 3 materials-15-02190-f003:**
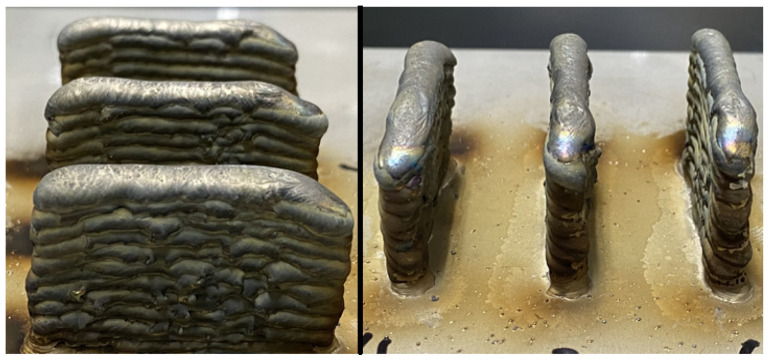
Example of Deposited Samples.

**Figure 4 materials-15-02190-f004:**
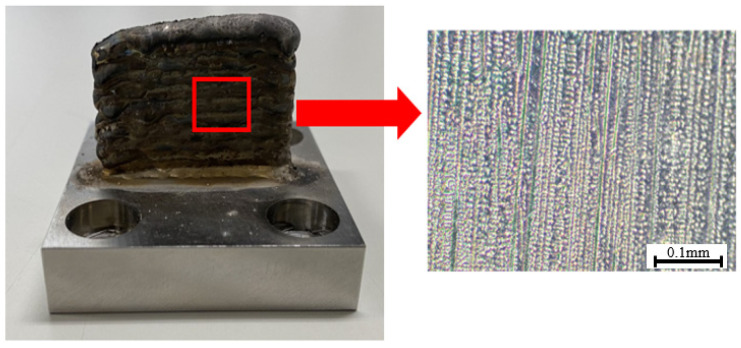
WAAM sample and dendrite structure representation.

**Figure 5 materials-15-02190-f005:**
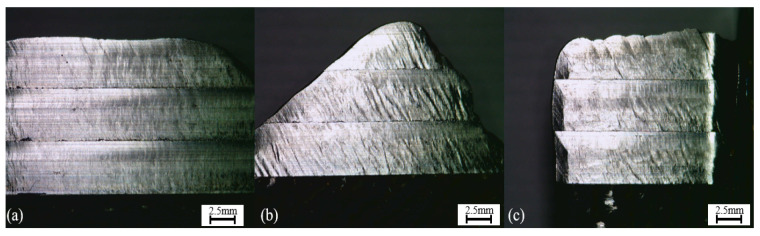
Machined Surfaces—(**a**) 0 degrees sample, (**b**) 35 degrees sample, (**c**) 90 degrees sample.

**Figure 6 materials-15-02190-f006:**
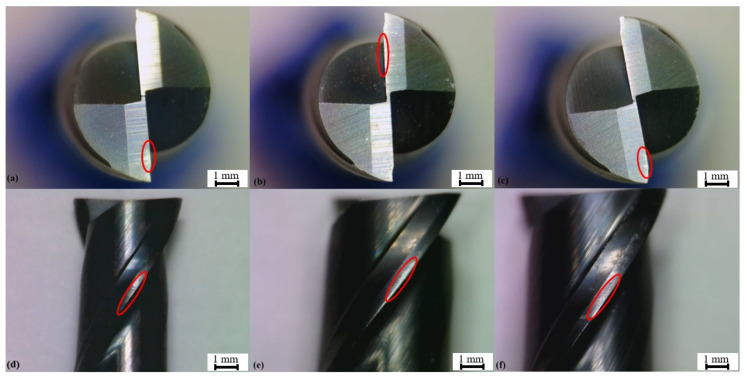
Tool Wear Examples—(**a**) 0 degrees, (**b**) 35 degrees, (**c**) 90 degrees top view sample tool wear. (**d**) Zero degrees, (**e**) 35 degrees, (**f**) 90 degrees side view sample tool wear.

**Figure 7 materials-15-02190-f007:**
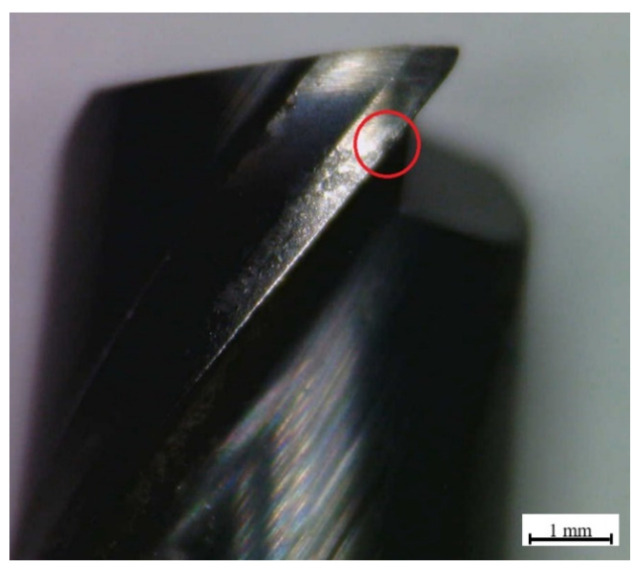
90-degree angle cutting tool-chipping.

**Figure 8 materials-15-02190-f008:**
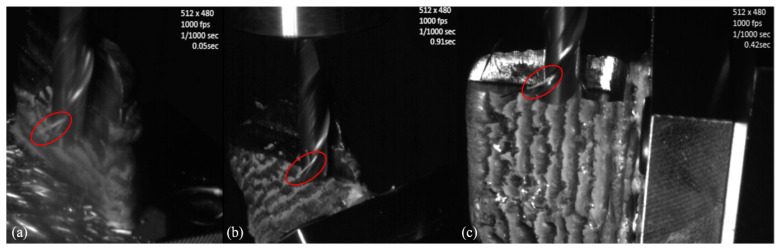
High-speed camera pictures—(**a**) 0 degrees, (**b**) 35 degrees, (**c**) 90 degrees.

**Figure 9 materials-15-02190-f009:**
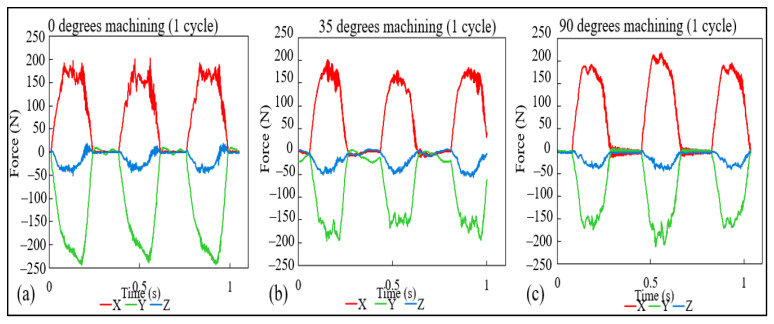
Cutting force measurement for one cutting cycle (**a**) 0 degrees, (**b**) 35 degrees, and (**c**) 90 degrees of cutting angle.

**Figure 10 materials-15-02190-f010:**
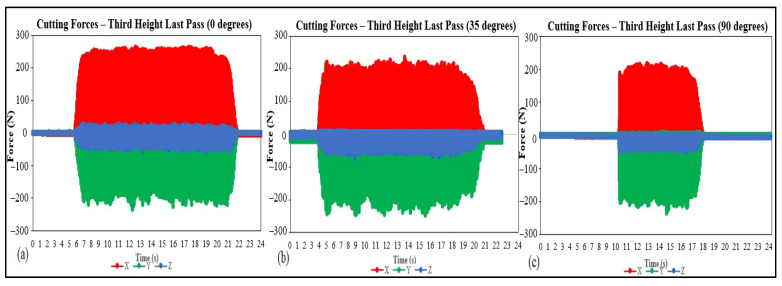
Cutting forces measurements for the last machining pass, (**a**) 0 degrees, (**b**) 35 degrees, and (**c**) 90 degrees of cutting angle.

**Figure 11 materials-15-02190-f011:**
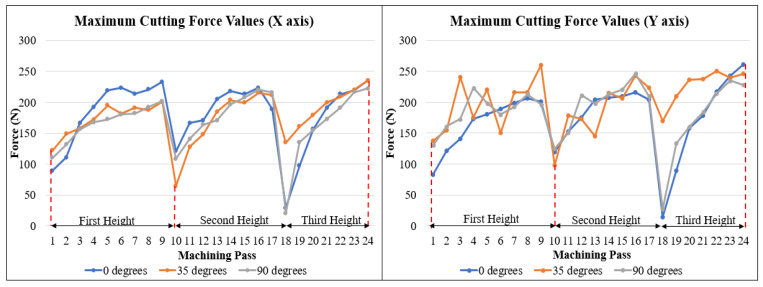
Maximum cutting force values for *x*- and *y*-axis.

**Figure 12 materials-15-02190-f012:**
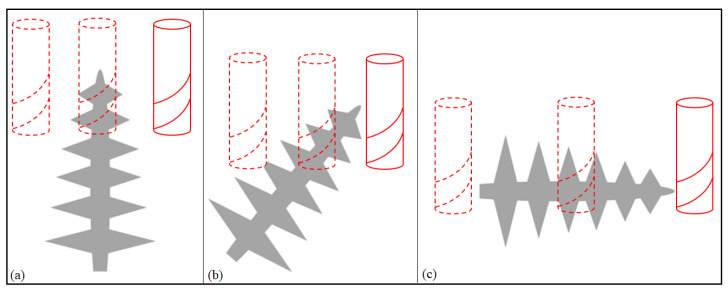
Machining Steps Representation (**a**) 0 degrees, (**b**) 35 degrees, (**c**) 90 degrees.

**Figure 13 materials-15-02190-f013:**
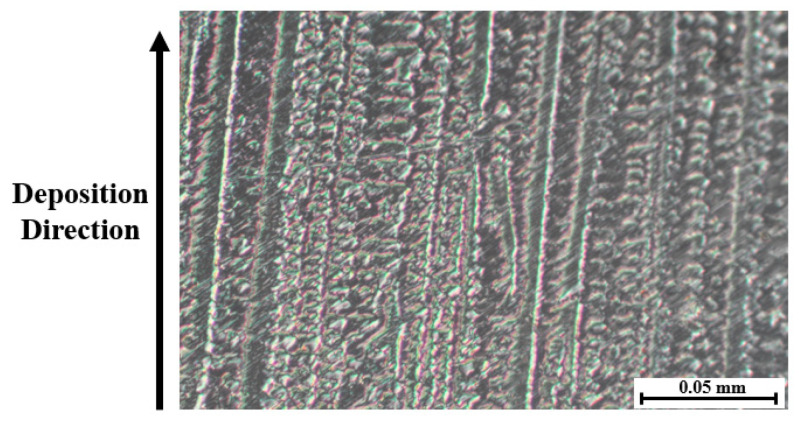
WAAM Inconel 718 microstructure.

**Figure 14 materials-15-02190-f014:**
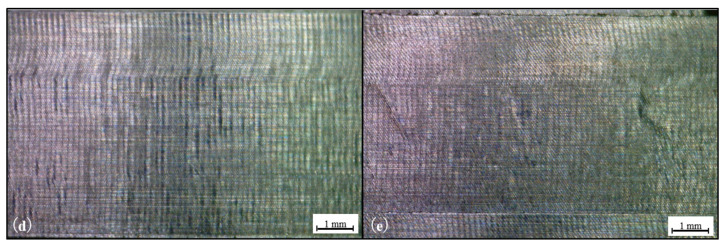
Machined Surfaces (Annealed sample) (**d**)—0 degrees, (**e**) 35 degrees.

**Figure 15 materials-15-02190-f015:**
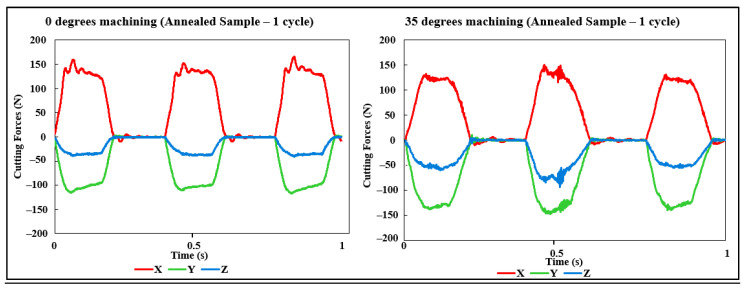
Cutting forces data—1 Cycle 0 and 35 degrees (Annealed Sample).

**Table 1 materials-15-02190-t001:** Sample Manufacturing Parameters (WAAM).

Sample Manufacturing Parameters (WAAM).							
Layer	Current	Voltage	Torch Speed	Wire Feed	Deposition Delay	Shielding Gas	Shielding Gas Flow Rate
1	119 A	13.5 V	100 cm/min	4.5 m/min	15 s	Argon	10 L/min
2–20	99 A	11.9 V	85 cm/min	3.7 m/min		Argon	10 L/min

**Table 2 materials-15-02190-t002:** Machining Parameters.

Machining Parameters				
Feed Rate	Spindle Rate	Cutting Speed	Axial Depth of Cut	Radial Depth of Cut
120 mm/min	800 rpm	15 m/min	0.3 mm	5 mm

**Table 3 materials-15-02190-t003:** Surface Roughness Measurements (Ra).

Surface Roughness Measurements (Ra)			
	(a)	(b)	(c)
Layer 1	0.56 µm	1.00 µm	0.82 µm
Layer 2	0.54 µm	1.02 µm	0.84 µm
Layer 3	0.54 µm	0.94 µm	0.73 µm

**Table 4 materials-15-02190-t004:** Surface Roughness Measurements (Rz).

Surface Roughness Measurements (Rz)			
	(a)	(b)	(c)
Layer 1	3.53 µm	5.52 µm	5.51 µm
Layer 2	3.43 µm	5.49 µm	5.27 µm
Layer 3	3.54 µm	5.07 µm	4.24 µm

**Table 5 materials-15-02190-t005:** Surface roughness average (Ra) cases (d) and (e).

Surface Roughness Average (Ra)	
(d)	(e)
0.36 µm	0.42 µm

**Table 6 materials-15-02190-t006:** Surface roughness Average (Rz) cases (d) and (e).

Surface Roughness Average (Rz)	
(d)	(e)
2.35 µm	2.86 µm

## Data Availability

The data presented in this study are available on request from the corresponding author.

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
