# Peer review of "Study on the Effects of Different Cutting Angles on the End-Milling of Wire and Arc Additive Manufacturing Inconel 718 Workpieces"

_materials, 2022, doi:10.3390/ma15062190_

Round 1

Reviewer 1 Report

This manuscript reports the effect of different cutting angles of the side milling on the machinability of Inconel 718 workpieces, which was produced by the Wire and Arc Additive Manufacturing (WAAM) technique. The effect of three different angles (0, 35 and 90) on the surface quality, tool wear, and cutting forces was analyzed and compared to each other.

Overall, this is a clear, concise, and well-written manuscript. The introduction is relevant and theory based. Sufficient information about the previous study findings is presented for readers to follow the present study rationale and procedures. The methods are generally appropriate. The results are clear. In my opinion, the manuscript is suitable for publication.

Best regards

Author Response

Dear Reviewer, 

We appreciate the attention and comments, 

We would also like to inform that according to all reviewers comments some improvements were made into the manuscript, some of the change points are enumerated below:

  • More Background information was added into the Introduction
  • Equipment data was further improved, with model, city, and country of origin added
  • Differences between WAAM and annealed samples further explained, with the addition of crystal sizes and deposition of laves phase
  • Enumeration of both topics and conclusion was reviewed
  • English was reviewed with more attention given to spelling, grammar and conciseness

We hope to meet the Journal Standards with the above mentioned changes

Kind Regards, 

Reviewer 2 Report

Authors attempted to present the good study on the cutting angles of AM Inconel 718; however, there is lot of scope to enhance the quality and contents of this manuscript. please observe the following suggestions:

1) In section 2: mention the model, make, city and country of all the equipments or instruments used int eh present study.

2) More Quantification data can be presented

Author Response

Dear Reviewer, 

We would Like to thank the attention, and the comments that were used for further improvement of the submitted manuscript, 

We would also like to inform that according to all reviewers comments some improvements were made into the manuscript, some of the change points are enumerated below:

  • More Background information was added into the Introduction
  • Equipment data was further improved, with model, city, and country of origin added
  • Differences between WAAM and annealed samples further explained, with the addition of crystal sizes and deposition of laves phase
  • Enumeration of both topics and conclusion was reviewed
  • English was reviewed with more attention given to spelling, grammar and conciseness

We hope to meet the Journal Standards with the above mentioned changes,

Kind Regards, 

Reviewer 3 Report

The present paper on the "Study on the effects of different Cutting angles on the End Milling of Wire and Arc Additive Manufacturing Inconel 718 Workpieces" describes the effects on the side milling of Inconel 718, which were manufactured by the Arc Additive Manufacturing Technique, due to varying cutting angles with regards to possible differences to the machinability. The analysis is supported by a very well described machining process focusing on the surface quality, the tool wear and cutting forces. Furthermore, the present results/findings are compared with cold rolled counterparts followed by a detailed description and discussion with regards to the cutting angles is provided by the authors.

All in all, there is nothing to criticize with this manuscript with regards to the description of the process, to the analysis of the present results or to the discussion including comparisons and benefit of the results for the science. The tables, graphs and images are illustrated in a satisfactory way.

The only one thing which needs to be improved/corrected is the spelling and the enumeration of the chapters (i.e. no chapter 4, after chapter 3 comes chapter 5) and statements in the discussion. The authors need to check and correct these small errors. Beyond these minor corrections, I consider this draft to be accepted.

Author Response

Dear Reviewer, 

We would like to thank the attention, and comments on our first version the manuscript, 

We would also like to inform that according to all reviewers comments some improvements were made into the manuscript, some of the change points are enumerated below:

  • More Background information was added into the Introduction
  • Equipment data was further improved, with model, city, and country of origin added
  • Differences between WAAM and annealed samples further explained, with the addition of crystal sizes and deposition of laves phase
  • Enumeration of both topics and conclusion was reviewed
  • English was reviewed with more attention given to spelling, grammar and conciseness

We hope to meet the Journal Standards with the above mentioned changes,

Kind Regards,

Reviewer 4 Report

The authors analyze the effects of cutting angles on Inconel 718 pieces fabricated by the WAAM technique. Surface roughness, tool wear, and cutting forces under different cutting angles are compared. The authors also analyze cold-rolled AMS 5662 alloy, which is composed of Inconel 718. The work is likely to be of interest to the community. The writing needs to be improved, and some items need to be clarified.

  1. The authors should make their language concise and clear. There are lots of redundant descriptions in the manuscript. Some sentences are very difficult to follow or understand. I strongly recommend the authors revise their writing.
  2. The introduction is not up to date. It doesn’t include enough background information.
  3. In Figure 10, the authors conclude that cutting force values are similar at three angles. However, it doesn’t seem so. The cutting forces at 0 degrees along X-axis is about 250 N, while at other two degrees are around 200 N. Such ~25% difference is not trivial. The larger force applied may explain the better surface quality at 0 degrees.
  4. If I understand correctly, the authors attribute the instability at 35 and 90 degrees to dendritic microstructure. It can explain the variation along X-axis, but cannot explain the inconsistent results along Y-axis.
  5. The differences between AM fabricated samples and cold-rolled samples are not limited to dendritic microstructure. Can the authors elaborate more differences, such as crystal size and phase?

Author Response

Dear Reviewer, 

We would like to thank the attention and the comments given, that could be used to further improve the manuscript, 

We would also like to inform that according to all reviewers comments some improvements were made into the manuscript, some of the change points are enumerated below:

  • More Background information was added into the Introduction
  • Equipment data was further improved, with model, city, and country of origin added
  • Incongruency regarding figure 10 cutting forces better explained, 
  • Differences between WAAM and annealed samples further explained, with the addition of crystal sizes and deposition of laves phase
  • Variations of the Y axis cutting forces hypothesized as not only being caused by the Anisotropy, but by the differences in the microstructure itself, like dendrite diameters,
  • Enumeration of both topics and conclusion was reviewed
  • English was reviewed with more attention given to spelling, grammar and conciseness

We hope to meet the Journal Standards with the above mentioned changes,

Kind Regards,

Round 2

Reviewer 4 Report

The issues raised have been well addressed. I recommend the revised manuscript for publication.